# Antibody Conjugation of Nanoparticles as Therapeutics for Breast Cancer Treatment

**DOI:** 10.3390/ijms21176018

**Published:** 2020-08-21

**Authors:** Alberto Juan, Francisco J. Cimas, Iván Bravo, Atanasio Pandiella, Alberto Ocaña, Carlos Alonso-Moreno

**Affiliations:** 1Oncología Traslacional, Unidad de Investigación del Complejo Hospitalario Universitario de Albacete, 02008 Albacete, Spain; alberto.juan@uclm.es (A.J.); franciscojose.cimas@uclm.es (F.J.C.); 2Centro Regional de Investigaciones Biomédicas, Unidad NanoCRIB, 02008 Albacete, Spain; ivan.bravo@uclm.es; 3Centro Regional de Investigaciones Biomédicas, Unidad Oncología Traslacional, 02071 Albacete, Spain; 4Centro de Investigación del Cáncer-CSIC, IBSAL- Salamanca and CIBERONC, 37007 Salamanca, Spain; atanasio@usal.es; 5Experimental Therapeutics Unit, Hospital clínico San Carlos, IdISSC and CIBERONC, 28040 Madrid, Spain; 6School of Pharmacy, University of Castilla-La Mancha, 02008 Albacete, Spain

**Keywords:** breast cancer, antibody drug conjugates, antibody conjugate nanoparticles, nanomedicines, targeted delivery systems

## Abstract

Breast cancer is the most common invasive tumor in women and the second leading cause of cancer-related death. Nanomedicine raises high expectations for millions of patients as it can provide better, more efficient, and affordable healthcare, and it has the potential to develop novel therapeutics for the treatment of solid tumors. In this regard, targeted therapies can be encapsulated into nanocarriers, and these nanovehicles are guided to the tumors through conjugation with antibodies—the so-called antibody-conjugated nanoparticles (ACNPs). ACNPs can preserve the chemical structure of drugs, deliver them in a controlled manner, and reduce toxicity. As certain breast cancer subtypes and indications have limited therapeutic options, this field provides hope for the future treatment of patients with difficult to treat breast cancers. In this review, we discuss the application of ACNPs for the treatment of this disease. Given the fact that ACNPs have shown clinical activity in this clinical setting, special emphasis on the role of the nanovehicles and their translation to the clinic is placed on the revision.

## 1. Introduction

Breast cancer is the second leading cause of cancer-related death and the most common invasive cancer in women. Classical available systemic therapies for the treatment of this disease include cytotoxic agents alone or in combination with targeted therapies [1,2]. However, the major limitations of systemic treatment include dose-limiting toxicity due to poor specificity, in addition to primary and secondary resistance to the given therapy. In this context, guided targeted therapies can reduce toxicity, improving the therapeutic index. At this moment, antibody–drug conjugates (ADCs) are the most successful targeted delivery systems [3,4,5,6]. To date, a total of eight ADCs have been approved by the United States Food and Drug Administration (FDA) (Figure 1). The translation of ADCs into clinically useful therapeutic options is still hampered by their construction as well as by the appearance of mechanisms of resistance [7].

Antibody-conjugated nanoparticles (ACNPs) are built on the potential of both antibody conjugation and nanotechnology [8,9]. In the same manner as ADCs, the membrane proteins expressed in tumoral cells can be used to design antibodies that are then conjugated, as a vector, to the nanoparticle (NP) [10]. In comparison to ADCs, ACNPs can deliver drugs in a controlled manner, preserving their chemical structure, avoiding unpredicted metabolization, and reducing toxicity.

This review focuses specifically on ACNPs under development for breast cancer therapy.

## 2. Selective Targeting of Breast Tumors

Nanoparticles (NPs) can offer several advantages as drug carriers, including those related to the nanoscale size, high surface/volume ratio, potential for selective targeting, and a controlled drug release [11,12,13]. It is considered that non-vectorized NPs of 100–400 nm diameter can accumulate within the tumor through the enhanced permeability and retention (EPR) effect [14,15]. They can deliver high concentrations of the drug to the site of interest by a convection and diffusion process [16] that can also reduce the effects to the surrounding tissues [17,18]. Even though the nanomedicine field had relied on the EPR effect to increase delivery to the tumor, recent works suggest potential limitations when explored in the clinic. Indeed, the EPR effect can differ among patients and types of tumors [19,20].

Vectorized NPs can be generated by conjugation with antibodies designed against membrane proteins expressed mainly on tumoral cells [21,22,23]. This ligand–receptor interaction induces internalization of the NPs via receptor-mediated endocytosis followed by drug release inside the cell [20,24]. Antibodies are the most frequently used ligands to actively target tumor cells due to their high specificity and affinity [20,25]. IgG is the most abundant antibody in normal human serum and the most widely used antibody to vectorize NPs. Smaller antibody fragments are also conjugated to improve tumor uptake [26,27]. Other options include the use of the antigen-binding fragments (Fab) generated by the enzymatic cleavage of a full-size antibody [28]. The use of antibodies to target the tumor and elicit independent therapeutic effects enhances the opportunities of ACNPs in clinic.

### 2.1. Targeting Breast Cancer with Antibody Conjugates

Ado-trastuzumab emtansine also known as T-DM1 (KadcylaTM; Genentech/Roche) is a human epidermal growth factor receptor 2 (HER2) ADC that comprises the humanized anti-HER2 IgG1 trastuzumab linked to the anti-mitotic agent emtansine, which is a tubulin polymerisation inhibitor that interferes with mitosis and promotes apoptosis. After binding to HER2, T-DM1 undergoes receptor-mediated internalization and lysosomal degradation, resulting in the intracellular release of DM1-containing cytotoxic catabolites. The binding of emtansine to tubulin disrupts microtubule formation during the mitotic process, resulting in cell-cycle arrest and apoptotic cell death. In vitro studies have also shown that, similar to trastuzumab, T-DM1 inhibits HER2-receptor signaling, mediates antibody-dependent cell-mediated cytotoxicity, and inhibits shedding of the HER2 extracellular domain in human breast cancer cells that overexpress HER2 [29].

Novel ADCs with different chemical properties have obtained clinical approval. For instance, the ADC DS-8201TM (trastuzumab deruxtecan) has been approved for HER2-positive metastatic breast cancer in patients receiving previous treatment with anti-HER2 therapies including T-DM1 [30]. Of note, this ADC has a cleavable linker inducing a bystander effect in the surrounding cells [31]. In this context, the development of this ADC is moving beyond breast cancer and, also, it includes patients with low to moderate levels of HER2.

Similarly, novel ACNPs can be active therapeutics. Figure 2 indicates the pros of ACNPs in comparison to ADCs. The presence of antibodies on the NPs surface can specifically bind to an overexpressed receptor on target cells, overcoming some of the limitations of nude NPs, including inefficient drug diffusion into the tumor and the induction of multiple-drug resistance mechanisms [32]. The ideal system for breast cancer treatment using ACNPs should control drug loading and delivery in an efficient manner.

After recognition and binding to the target antigen, the internalization of the ADC-antigen complex into the cell is produced through receptor-mediated endocytosis [5,33]. Internalization results in early endosomes formation [34]. The high affinity of the antigen–antibody binding stabilizes the interaction, preventing this back circulation and enhancing the specificity of this therapeutic approach [35]. Finally, the early endosome is transformed into a late endosome by reducing the presence of proteins involved in recycling. This late endosome couples to lysosomes that cleave the ADC, which subsequently release the free cytotoxic warheads into the cytoplasm [34,36], interfering with the cellular mechanisms and ultimately promoting cell death [37,38]. Figure 3 shows the action mechanism of ADCs in comparison with that proposed for ACNPs.

### 2.2. Conjugation Strategies for ACNPs Generation

NPs can be functionalized with antibodies or antibody fragments by adsorption or covalent binding. The immobilization of antibodies should ensure the desired amount of these biomolecules per nanoparticle and their correct orientation [39]. The higher the ratio of antibody molecules over the NP surface, the lower the spatial accessibility of the antigen that is present [40]. Moreover, the coupling method must maintain the biological activity of the antibody [41].

Adsorption is a non-covalent immobilization strategy that includes physical adsorption and ionic binding [42]. Physical adsorption involves antibody attachment to the NP surface through weak interactions (electrostatic, hydrogen binding, hydrophobic and van der Waals attractive forces) [43], while ionic binding is based on ionic linkages between oppositely charged surfaces of the antibody and the NPs [44].

Covalent binding requires prior activation of the nanoparticle [45]. The most common covalent methods are based on carbodiimide chemistry, maleimide chemistry, or “click chemistry”. Carbodiimide chemistry is a simple method and chemical modification of the antibody is not required. However, coupling between the functional groups and cross-linkers is not selective and leads to the major disadvantage of lacking control over antibody orientation onto the nanoparticle surface [46,47]. Maleimide chemistry involves binding through sulfhydryl groups of antibodies. These chemical groups are not as abundant as primary amines in the antibody structure [48,49]; thus, the incorporation of free sulfhydryl groups is required [50,51]. The use of heterobifunctional maleimide cross-linkers provides greater flexibility to the conjugation and control over the reactions in terms of cross-linking sites and extent [52,53]. Maleimide reactions involve free amino groups present at the N-terminal end of a protein or in lysines. However, non-selectivity in bio-conjugation to cysteines due to exchange reactions with thiol-containing proteins in serum has been reported. “Click chemistry” chemical reactions provide orthogonality, site-specificity, and a favorable reaction rate. Besides, the reactions are performed with ease and require no or minimal purification [54,55,56,57].

Most covalent strategies produce low coupling efficiency and randomly oriented antibodies. Non-covalent approaches using adapter biomolecules can provide orientation of the immobilized antibodies on the NP surface [58]. The most relevant binding strategy with adapter molecules exploits biotin–avidin interaction as the strongest non-covalent biological interaction between a protein and a ligand [59]. The most common approach using biotin–avidin interaction implies chemical modification of the antibody with biotin (biotinylation) and functionalization of the nanoparticle with avidin or its derivatives [60].

### 2.3. Remaining Challenges for Bringing ACNPs to the Clinic

ACNPs have failure in clinical translation. The lack of knowledge about the interaction between nanocarriers and biological systems, poor tumor accumulation, inadequate pharmacokinetics, the safety issue of raw materials for NPs generation, and limited number of reported in vivo studies remain limitations to upgrade ACNP to the clinic [61]. In general terms, there is not sufficient understanding about the interaction between NPs with biological macrostructures—even more so when referring to ACNPs. For example, it is well known that the surface of the non-targeted NPs is quickly covered by serum proteins. This fact implies important changes in NPs stability and metabolism that cannot be merely anticipated when using in vitro studies. In this matter, the influence of the size, shape, and surface charge of ACNPs are crucial to understand immune response and therefore facilitate better ACNPs design [62]. At this moment, the influence of the shape is a much-discussed subject and is still at an early stage. Further investigations and in vivo outcomes are required to determine the effects of NPs shape, size, and surface charge on cellular uptake [63].

On the other hand, NPs must prevent the mononuclear phagocyte system to increase circulating time. Toxicity, immunogenicity, and mechanism of action studies support the grafting of polyethylene glycol (PEG) to the nanoparticle surface as an adequate strategy for cellular internalization [64]. Other alternatives have been explored such as the use of protein and cell membrane coatings [65,66]. Enhanced efforts are underway to develop reliable technologies in this matter. Again, deeper in vivo outcomes—or, where that is not possible, simulated in vivo model culture systems—to mimic the specific tumor microenvironment are required. Insufficient accumulation in the tumor is another concern for clinical translation. As a very small quantity of NPs are delivered to a solid tumor [67], a superior tumor accumulation with ACNPs is expected. Some recent clinical trials point in that way, improving the overall patient survivals [68]. Promising approaches to the controlled release of the drug via an external stimulus are being pursued, but there is very scarce knowledge regarding ACNPs.

The choice of the raw material for ACNPs generation is dependent on the structure of the cargo to ensure high entrapment. Clinical safety is another issue for NPs entering clinical trials. Preclinical studies concerning the stability, sterility, and in vivo cytotoxicity and immunotoxicology are always required before entering the clinical phase. However, despite formal toxicology evaluation of the raw materials, the toxicity of NPs related clinical failures is observed [69]. Still, researchers must understand the chemistry of NPs to better design ACNPs. It seems that the use of biodegradable and biocompatible polymers might reinforce the potential translation of ACNPs to the clinic.

A need for better animal models that could predict toxicity and efficacy in humans is a main goal particularly for those agents with immunologic properties. On the other hand, ACNPs offer further development opportunities. (1) First, there is the payload; in contrast to ADCs, a wide variety of drugs can be incorporated into ACNPs. More importantly, a direct linker to the drug is not required, avoiding changes in the chemical structure that could modify the antitumoral properties. (2) Second, there is the drug-to-antibody ratio; the versatility of the ACNPs to modulate the cargo of antibodies over the surface by different conjugation strategies may ensure the internalization of a much higher concentration of the drug [70]. (3) Third, there is the drug release; the release of the drug in ACNPs is independent of the linker. The drug release from ACNPs is only a consequence of drug diffusion and nanoparticle degradation [71]. (4) Last, there are the multivalent effects; the conjugation of antibodies over the surface of NPs can provide therapeutic effects also.

## 3. ACNPs for Breast Cancer Therapy

There are different nanocarriers reported for the generation of ACNPs as novel therapeutics. Among them, inorganic, polymeric ACNPs, and immunoliposomes have been the most evaluated ACNPs in this setting [72].

### 3.1. Immunoliposomes

Since the discovery of Doxil^TM^ [73], a liposomal-based nanocarrier for doxorubicin (DOX) currently used to treat metastatic breast cancer, liposomes have been the most clinically successful nanocarriers for the treatment of cancer [74]. Doxil^TM^, Myocet^TM^, and Lipusu^TM^ are all liposomal formulations of different chemotherapies approved for breast cancer therapy. Myocet^TM^ is a DOX-loaded liposomal formulation approved for metastatic breast cancer with reduced cardiotoxicity compared to traditional DOX [75]. Lipusu^TM^ represents the first paclitaxel (PTX) liposome formulation, which was approved in China in 2003 for clinical use. Lipusu^TM^ showed similar clinical activity in breast cancer, but with lower side effects compared to PTX [76,77]. Finally, the liposomal cytarabine DepoCyt^TM^ is in the Phase III clinical stage for the treatment of leptomeningeal metastasis from breast cancer [78].

Immunoliposomes are liposomal formulations with antibody molecules conjugated to the surface. There are numerous types of targeting ligands reported in the literature for the generation of targeted liposomes (Table 1) [79]. To date, there have been five targeted liposomes in clinical trials: Endotag-1^TM^, C225-ILs^TM^, MM-310^TM^, MM-302^TM^, and MBP-42^TM^ [61]. C225-ILs^TM^, MM-302^TM^, and MM-310^TM^ are immunoliposomes. C225-ILs^TM^ are under clinical evaluation in a Phase I Clinical trial for glioblastoma (NCT03603379). MM-302^TM^ and MM-310^TM^ are immunoliposomes for the treatment of breast cancer. MM-302^TM^ reported negative outcomes in clinical trials in 2016 (NCT02735798). MM-310^TM^ reported cumulative peripheral neuropathy in a Phase I clinical trial, and its development was terminated (NCT03076372).

Looking at the scientific literature, most studies used trastuzumab as the antibody for conjugation with the aim of targeting HER2 overexpressing tumors (Table 1). Of note, other antibodies have been used for conjugation with immunoliposomes in an intent to overcome drug resistance mechanisms [87,88,89,104]. The review from Benz et al. in 2000 claimed that approaches involving clinical testing in vivo with advanced HER2 overexpressing breast cancer are urgently needed to provide conclusive evidence for the superior therapeutic efficacy of anti-HER2 immunoliposomes [86]. In this context, Moase et al. reported DOX-loaded trastuzumab effective immunoliposomes in treating early lesions in pseudometastatic and metastatic mice models, but limitations to the access of the targeted liposomes to tumor cells in the primary tumor compromised their therapeutic efficacy in treating the more advanced lesions [83]. Therapeutic efficacy studies in vivo showed that immunoliposomes constructed with different fragments derived from trastuzumab are significantly superior to free DOX, DOX-loaded liposomes, and DOX-loaded trastuzumab immunoliposomes [81,82,84,105]. However, concerns were raised regarding the real mechanism of action [106]. In 2016, shortly before the negative clinical outcomes of MM-302^TM^ for HER2-positive metastatic breast cancer, researchers reported a novel combination therapy that efficiently targets HER2-overexpressing [80]. The mechanism of action of MM-302^TM^ was not altered by the presence of trastuzumab, while trastuzumab decreased intracellular signaling m × (*p* − Akt).

Other agents apart from DOX have also been included in ACNPs to treat breast cancer [73,74,75,76,77,78]. Similar to the use of DOX, taxanes have demonstrated significant toxicity in normal tissues for breast cancer therapy. The patient’s quality of life is significantly impacted by side effects. In this context, immunoliposomes offer substantial advantages compared to common liposomes and ADCs. An evaluation of the antitumor activity and mechanism of action of PTX-loaded trastuzumab immunoliposomes were performed in various breast cancer cells and in xenograft nude mouse models [92]. Immunoliposomes showed superior antitumor efficacy and higher tumor tissue distribution of PTX in the BT-474 xenograft model compared to Taxol^TM^ and non-targeted liposomes. However, in the MDA-MB-231 xenograft model, PTX-loaded liposomes and immunoliposomes showed similar tumor outcomes. In vitro studies reported by Fanciullino et al. showed a higher antiproliferative efficacy of docetaxel (DTX)-loaded trastuzumab immunoliposomes in breast cancer cell lines than a standard combination of DTX plus trastuzumab [93]. One year later, this group questioned the use of anti-HER2 antibodies to improve liposomes distribution and efficacy, reporting no difference in tumor uptake between immunoliposomes and standard DTX liposomes [94]. Very recently, they optimized the DTX-loaded immunoliposomes using a quantitative assay based on flow cytometry to demonstrate that the density of the targeting agent should be finely tuned to get the highest efficacy [95].

The mammalian target of rapamycin pathway plays a key role in the malignant progression of breast tumor cells [107]. Consequently, rapamycin has been extensively studied as an option for breast cancer treatment [108]. The co-delivery of PTX and rapamycin from trastuzumab-targeted immunoliposomes reduced tumor growth in vivo compared to untreated controls [79,86]. On the other hand, the co-delivery of curcumin and resveratrol from trastuzumab-targeted immunoliposomes were also explored for HER2 breast cancer [100]. The combinations of the two compounds in their free form did not improve the cytotoxic effect, but the co-loaded immunoliposomes significantly increased the cytotoxic effect in MCF-7 and JIN cell lines.

A considerable body of clinical trials assessing the putative benefit of statins to impair proliferation on breast cancer cells have been performed [109]. The mechanism for this effect remains poorly understood and requires further investigation. Furthermore, there are still no effective and safe methods to provide statins at doses effective in breast cancer treatment, which is mostly due to their lipophilic character and poor bioavailability. To solve this problem, the encapsulation of simvastatin on epidermal growth factor receptor (EGFR)- and HER2-targeted immunoliposomes were proposed [96,97]. In vitro and in vivo studies showed the effectiveness of the immunoliposomes in the induction of apoptosis.

Better therapeutic outcomes are expected for the use of gemcitabine and bleomycin in breast cancer therapy. Bleomycin is highly cytotoxic when delivered directly to the cytoplasm but relatively innocuous extracellularly, whereas gemcitabine is highly hepatotoxic. Trastuzumab-targeted immunoliposomes were used as nanocarriers to overcome such limitations. Although further in vivo studies are required for its clinical evaluation, gemcitabine-loaded immunoliposomes allowed decreasing the concentration-dependent antitumoral activity for gemcitabine therapy [102]. The direct linking of bleomycin-loaded immunoliposomes to the pore-forming protein listeriolysin O was the strategy proposed to allow the liposomal cargo of bleomycin to pass into the cytoplasm [101].

Finally, target-specific gene delivery to HER2 and epidermal growth factor receptor (EGFR)-overexpressing cells were successfully developed by the insertion of lipid-modified anti-HER2-Fab or anti-EGFR Fab into the preformed liposomes by conventional maleimide conjugation [103]. Recently, hybrid immunoliposomes encapsulating a Poly (L-lysine)-siRNA complex were designed to silence the epithelial cell adhesion molecules highly expressed in breast cancer [104].

The recent advancements involving immunoliposomes to target breast cancer were shortly reviewed by Khan et al. in 2018 [105]. Since then, researchers have reported combinatorial strategies involving immunoliposomes [85,91]. Identifying appropriate biomarkers for patient stratification will be of high importance in future trial design. However, advanced antibody engineering and innovative manufacturing techniques must be addressed for the generation of more effective immunoliposomes. Despite the significant progress in the liposome field, there are not any approved immunoliposomes on the market.

### 3.2. Inorganic ACNPs

Inorganic NPs have been classified into superparamagnetic NPs, quantum dots, quantum rods, nanoshells, silica nanoparticles, gold nanoparticles, and nanocages [110]. Some inorganic ACNPs have been developed as diagnostic biomarkers to be used as screening tools [110,111,112] and for hyperthermia therapy [113,114,115,116,117,118,119]. To a lesser extent, some of them have been explored for the tumor-selective delivery of chemotherapeutics (Table 2) [110].

There are several inorganic NPs in clinical trials; all of them are metallic NPs [61]. Nanotherm^TM^ therapy is based on injecting iron oxide nanoparticles directly into the tumor [144]. Phase III clinical trials for the hafnium oxide NPs NBTX3^TM^ indicated for head and neck cancer or non-small cell lung cancer are ongoing. Other inorganic NPs are currently in early stages of clinical development in different indications [145,146,147].

To date, inorganic ACNPs have been used for early detection, thermotherapy, and biomarker identification for the treatment of breast cancer (Table 2). Moreover, iron oxide superparamagnetic (SPIONs)-, gold- and quantum dots- (QDs)-based ACNPs have been proposed as drug and siRNA delivery systems [25,148,149]. In this regard, the strategy relies on an organic polymeric coating on the NPs surface to enable the encapsulation of the drug or the siRNA.

In a first approach, SPIONs and DOX were encapsulated into a poly (lactic-co-glycolic acid) (PLGA) core through a nano-emulsion method followed by trastuzumab conjugation by EDC/NHS chemistry [92]. The ACNPs showed excellent colloidal stability in aqueous phase and released DOX sustainably. However, the ACNPs exhibited the saturation magnetization superparamagnetic behavior due to the presence of organic components such as DOX, poly (lactic-co-glycolic acid) (PLGA) and polyvinyl alcohol (PVA).

Regarding SPIONs-based ACNPs, PTX, DOX, and siRNA were successfully encapsulated in ACNPs, showing selective breast cancer cell death [137,138,139]. A polyethylenimine coating allowed the generation of anti-HER2 ACNPs for siRNA delivery [138]. The ACNPs showed intracellular delivery and therapeutic effects of vascular endothelial growth factor (VEGF) siRNA against cancer cells. DOX were encapsulated in dextrane-modified SPIONs by ionic gelation and HER2 monoclonal antibody conjugated via EDC/NHS chemistry over the surface of the NPs [139]. 3-(4,5-Dimethylthiazol-2-yl)-2,5-diphenyltetrazolium bromidefor (MTTs) assays and transmission microscopy confirmed the selective uptake and cellular internalization of the DOX-loaded ACNPs. Interestingly, a multifunctional nanoplatform for DOX delivery and positron emission tomography-magnetic (PET/MR) imaging were obtained by loading DOX into ^64^Cu-labeled trastuzumab and rituximab ACNPs [140]. The simultaneous co-delivery of DOX and PTX from magnetic trastuzumab-conjugated ACNPs was reported to successfully suppress cancer growth in vivo [141].

Drug delivery using QDs-based ACNPs is very scarce. To highlight, the work of Zhang et al. reported the delivery of HER2 siRNA to overexpressing SKBR3 breast cancer cells through ACNPs conjugated with HER2 antibodies over a modified chitosan surface modification [150].

In the case of gold ACNPs, anti-HER2 conjugated gold ACNPs as a theranostic probe for imaging and breast cancer treatment have been described [113]. Strong therapeutic effects were reported using hybrid Au-Fe3O4 nanoparticles conjugated with trastuzumab for the delivery of cisplatin [142]. Recently, anti-Wnt-1 monoclonal antibodies were conjugated to inorganic NPs to induce apoptosis without requiring a therapeutic payload [143,151].

### 3.3. Polymeric ACNPs

Polymeric ACNPs are generated from biocompatible and biodegradable polymers, using either natural or synthetic materials. The delivery of the drug is based on a triphasic profile where there is a first step called “burst release”, followed by a second diffusion step that finishes with a third process called the erosion stage [152]. The customized design of the polymeric structure can control the extension of each one of these steps and therefore optimize the delivery of the compound.

The targeting ligands of immunoliposomes are commonly attached by unspecific chemical conjugation, bearing risks derived from structural heterogeneity. The first studies on non-liposomes formulation that demonstrate the specific targeting of anti-HER2 ACNPs were published in 2004 by Langer et al. [153,154]. These data provided the basis for the development of stable and biological active polymeric ACNPs for breast cancer therapy.

From the first FDA approval of Doxil^TM^ to the latest European Medicines Agency (EMA) approval of Apealea^TM^, there are at least 15 cancer nanomedicines on the market. None of them are ACNPs [122,123,124]. Regarding polymeric nanomedicines approved for breast cancer therapy, Genexol-PM^TM^ is a PTX polymeric micelle formulation that has been clinically approved to treat breast cancer in South Korea in 2007 [155,156,157]. PICN is a 100–110 nm formulation of PTX stabilized with polymer and lipids and was approved in India for metastatic breast cancer in 2014 [158].

In 2013, 13 targeted NPs had progressed into clinical trials, but their therapeutic efficacy in humans has not been proven yet. Among them, 2 were of polymeric nature [61]: BIND-014^TM^, composed of a copolymer polylactide–polyethylene glycol (PLA-PEG) for the controlled release of DTX against cancer (NCT01300533, NCT02479178, NCT02283320, NCT01792479, NCT01812746) [159,160,161], and CALAA-01TM, which were targeted polymeric NPs generated for the siRNA-mediated treatment of solid tumors (NCT00689065) [162]. Of note, no polymeric ACNPs have entered clinical studies.

Table 3 compiles polymeric ACNPs reported for the treatment of breast cancer. As a proof of concept, DOX was encapsulated into a PLGA core through nanoemulsion methods followed by trastuzumab conjugation [163,164,165]. Trastuzumab-targeted PLGA ACNPs were also successfully designed to entrap rapamycin 150 and DTX [133].

Knowing that the combination of tamoxifen with trastuzumab promoted therapeutic efficacy in treating HER2-positive and ER-positive metastatic breast cancers, PLGA ACNPs were generated by polyvinyl-pyrrolidone coating and the subsequent conjugation of trastuzumab by EDC activation [168]. In vivo studies showed an inhibition rate of ACNPs higher than the non-targeted NPs and free tamoxifen. The control of the surface density of trastuzumab over PTX-loaded PLGA NPs showed impact over the performance of the ACNPs [170].

In 2016, Chudasama et al. highlighted how antibody-derived fragments can been used for ACNPs generation [28]. They claimed that the conjugation of antibody-derived fragments is a step in the right direction to overcome fundamental and practical issues encountered during ACNPs generation. However, advancements in protein engineering and expression are still needed for a careful and precise disassembly of a full antibody easily. One representative work with polymeric ACNPs is the selective insertion of pyridazinedione moieties bearing reactive handles into reduced disulfide bonds to site-selectively modify trastuzumab [181].

Due to many patients acquiring trastuzumab resistance, pertuzumab, a monoclonal antibody that binds to HER2 and sterically blocks the homodimerization and heterodimerization of HER receptors, was presented as a potential vector for ACNPs generation. Nevertheless, no ACNPs have been generated to date based on this target ligand [182]. To increase the bioavailability and reduce the innate immunogenicity of trastuzumab coumarin-loaded ACNPs, a PEG coating over the targeted surface of the ACNPs was achieved [174]. PEG-modified ANCPs showed the most optimal performance in terms of a reduction in phagocyte uptake as well as immunogenicity.

For the first time, redox responsive-modified copolymers of PLA and PEG were designed as raw materials for ACNPs generation [171]. By comparing in vivo and in vitro targeting efficiency, there were not differences observed regarding their previous work focused on polycaprolactone ACNPs [183]. Epirubicin-loaded PLGA ACNPs were obtained by nanoprecipitation methods and trastuzumab conjugated with the help of carbodiimide chemistry to modify the pharmacokinetic parameters and the therapeutic index of the new compound [176]. On the other hand, DOX/cisplatin co-loaded chitosan ACNPs were designed with the aim of obtaining a synergistic interaction [177]. The influence of the ACNP shapes on efficacy is still under evaluation and needs further investigation [184]. Worn-like polycaprolactone (PCL)-PEG ACNPs for the controlled release of PTX to HER2 positive breast cancer cells were designed to enhance the binding capability of the nanoparticles [175].

Recently, our group reported the development of dasatinib-loaded ACNPs. The ACNPs were generated by nanoprecipitation methods and trastuzumab anchored after the polyethylenImine (PEI) coating using carbodiimide chemistry. The results showed efficacy, particularly in HER2-overexpressing cells, maintaining the same mechanism of action as dasatinib given alone [180].

## 4. Outlook and Recent Implications in Breast Cancer Therapy

There is no ACNP that has yet reached clinical stage, and only non-vectorized NPs have entered the clinical setting. While marketed NPs have shown a more favorable pharmacokinetic profile than their free payload, the optimization of aspects such as nanoparticle size, shape, and surface charge should be taken into consideration to improve efficacy. To reach this goal, tumor type and location should be considered due to the specific microenvironment characteristics.

In an effort to increase tumor delivery, several approaches are now emerging to augment the permeability and penetration of the particle within the tumor. Those include the selective targeting of components of the tumor neovasculature or targeting tumoral cells to enhance the immune system. Therefore, ACNPs can be oriented not only against the tumoral component but against the extracellular compartment, the vasculature structure, or the immune system. In this way, using antibodies to target negative immune modulators can augment the efficacy of the payload included in the NPs, inducing a double effect: the one produced by the targeted agent or chemotherapy payload, and the one against the specific antibody. An example would be the use of anti-PD-L1 (programmed death-ligand 1) antibodies that disrupt the inhibitory effect of the PD1/PD-L1 axis by acting on this receptor expressed in tumoral cells. A similar approach using ADCs has been recently reported [185].

ADCs have proven their ability to deliver cytotoxic payloads to tumors and are currently the most beneficial targeted, conjugated therapy for patients. ACNPs allow for existing chemotherapies to be made available in nanomedicine preparations. Drug release from NPs can be more finely controlled with a range of nanoparticle materials and co-excipients. Selection of the more adequate ones could make a difference. Since ACNPs favors drug structure preservation, ACNPs may provide benefit over ADCs when targeting receptors that have also a biological effect and by inducing a bystander effect. This effect can be particularly relevant in HER2-positive tumors that become resistant to TDM1 due the proliferation of breast cancer cells with low levels of HER2. As mentioned, in the age of immunotherapy, the ability to use ACNPs to augment immune responses in tumors holds much promise. In this context, given the fact that immunotherapy has shown efficacy in triple negative breast cancer, evaluation of agents acting in this indication is a main goal [186].

In conclusion, with strategies enhancing the ability of these agents to reach tumors by facilitating active targeting, combined with improved uniform manufacturability using conjugation chemistries, it is anticipated that there will be an increase in the interest of this family of agents for clinical development.

## Figures and Tables

**Figure 1 ijms-21-06018-f001:**
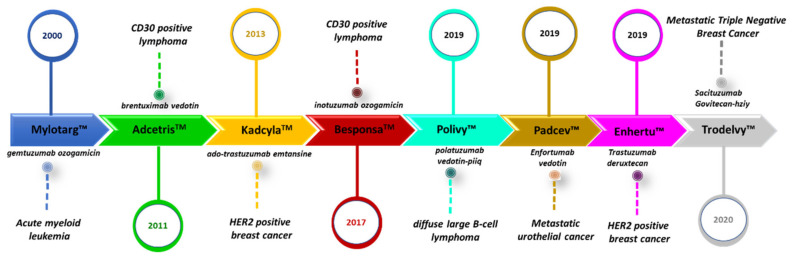
Antibody-drug conjugates (ADCs) approved by the Food and Drug Administration (FDA).

**Figure 2 ijms-21-06018-f002:**
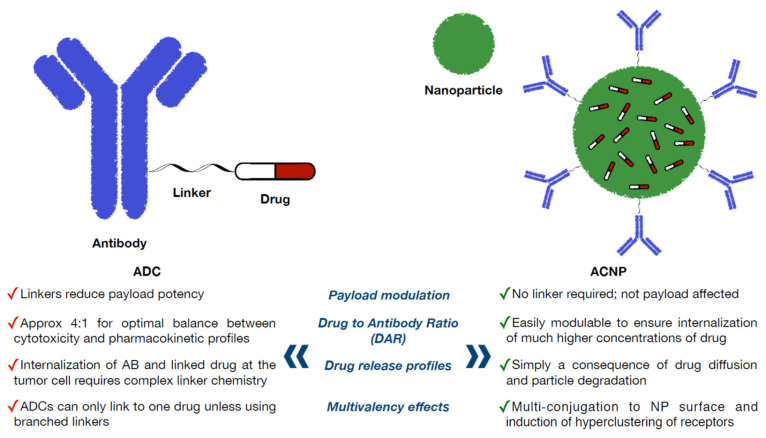
Pros of ACNPs in comparison to ADCs.

**Figure 3 ijms-21-06018-f003:**
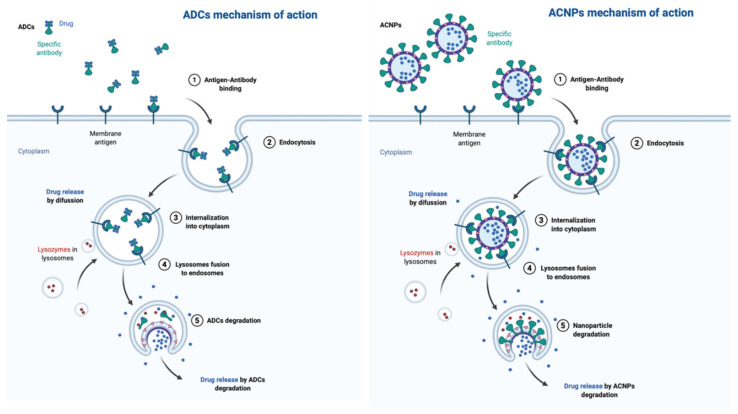
Action mechanism of antibody-conjugated nanoparticles (ACNPs) and ADCs.

**Table 1 ijms-21-06018-t001:** Immunoliposomes for breast cancer therapy. DOX: doxorubicin, HER2: human epidermal growth factor receptor 2, PTX: paclitaxel, HBEGF: Heparin binding EGF like growth factor, EGFR: Epidermal Growth Factor Receptor, RON: Recepteur d’Origine Nantais, DTX: Docetaxel, EpCAM:Epithelial Cell Adhesion Molecule.

Drug	Tumor Antigen	Status	Reference
DOX	HER2	In vitro	[80,81,82]
DOX	HER2	In vitro/In vivo	[83,84,85]
DOX	EGFR	In vitro/In vivo	[86]
DOX	HBEGF	In vitro/In vivo	[87]
DOX	RON	In vitro	[88]
DOX	ErbB2	In vitro/In vivo	[89]
DOX	HER2/CD3	In vitro	[90]
PTX	HER2	In vitro/In vivo	[91]
DTX	HER2	In vitro	[92]
DTX	HER2	In vitro/In vivo	[93]
DTX/Ephrin A2	HER2/HER2	In vitro	[94]
Simvastatin	HER	In vitro	[95]
Simvastatin	EGFR	In vitro/In vivo	[96]
Rapamycin/rapamycin-PTX	HER	In vitro/In vivo	[97,98]
Curcumin-reverastrol	HER2	In vitro	[99]
Bleomycin	HER2	In vitro	[100]
Gemcitabine	HER2	In vitro	[101]
siRNA	EGFR	In vitro/In vivo	[102]
siRNA	EpCAM	In vitro/In vivo	[103]

**Table 2 ijms-21-06018-t002:** Inorganic ACNPs for breast cancer diagnosis and therapy. QDs: quantum dots, SPIONS: iron oxide superparamagnetic, VEGF: vascular endothelial growth factor, mTOR: mammalian target of rapamycin, ER: estrogen receptor, PR: progesterone receptor, Wnt-1: protein that in humans is encoded by the Wnt1 gene, CD: cluster of differentiation, TMUC1: polymorphic epithelial mucin.

**Early detection**				
**NPs**	**Tumor antigen**		**Status**	**Reference**
SPIONs	HER2		In vitro/In vivo	[120,121,122,123]
SPIONs	EGFR		In vitro	[124]
SPIONs	VEGF		In vitro/In vivo	[125]
Manganese oxide	CD10539		In vitro/In vivo	[126]
Mesoporous	TMUC1		In vitro/In vivo	[127,128]
**Thermotherapy**				
**NPs**	**Tumor antigen**		**Status**	**Reference**
Gold	HER2		In vitro	[113]
Gold	EGFR		In vitro/In vivo	[114]
Bismuth-mesoporous	HER2		In vitro/In vivo	[115]
Gold nanocages	EGFR		In vitro	[116]
Silica-gold nanoshells	HER2		In vitro	[117]
SPIONs	HER2		In vitro/In vivo	[118]
Gold Nanoantenna	HER2		In vitro	[119]
**Biomolecular profiling**				
**NPs**	**Tumor antigen**		**Status**	**Reference**
QDs	HER2		In vitro/In vivo	[129,130,131,132,133]
QDs	HER2/ER		In vitro	[134]
QDs	EGFR		In vitro	[135]
QDs	HER2/ER/PR/mTOR/EGFR		In vitro	[136]
**Drug Delivery**				
**Drug**	**NPs**	**Tumor antigen**	**Status**	**Reference**
PTX	SPIONs	HER2	In vitro/In vivo	[137]
siRNA	SPIONs	HER2	In vitro	[138]
DOX	SPIONs	HER2/VEGF	In vitro	[139]
DOX	SPIONs	HER2	In vitro	[140]
DOX–PTX	SPIONs	HER2	In vitro/In vivo	[141]
siRNA	QDs	HER2	In vitro	[134]
Cisplatin	Au-Fe_3_O_4_	HER2	In vitro	[142]
None	Gold	Wnt-1	In vitro	[143]

**Table 3 ijms-21-06018-t003:** Polymeric ACNPs for breast cancer therapy. PEI: polyethylenImine, PLA-PEG: polylactide-polyethylene glycol, PLGA: poly(lactic-co-glycolic acid).

Drug	Polymer	Tumor Antigen	Status	Reference
None	PLGA	HER2	In vitro	[163]
DOX	Poly(TMCC-co-LA)-g-PEG-furan	HER2	In vitro	[165]
Tamoxifen	PLGA	HER2	In vitro	[164]
Rapamycin	PLGA	EGFR	In vitro	[166]
DTX	PLA-PEG	HER2	In vitro	[167]
Tamoxifen	PLGA	HER2	In vitro/In vivo	[168]
Curcumin	PLGA	AnxA2	In vitro/In vivo	[169]
PTX	PLGA	HER2	In vitro	[170]
DOX	PLA-PEG	HER2	In vitro/In vivo	[171]
DOX	chitosan	HER2	In vitro	[172]
DOX	PCL-PEG-PCL-urethane	HER2	In vitro/In vivo	[173]
Coumarin	PLA-PEG	HER2	In vitro	[174]
PTX	PCL-PEG	HER2	In vitro	[175]
Epirubicin	PLGA	HER2	In vitro	[176]
DOX–cisplatin	Chitosan	HER2	In vitro	[177]
siRNA	PEI-PEG	HER2	In vitro	[178]
PTX	PLGA	HER2	In vitro/In vivo	[179]
Dasatinib	PLA-PEI	HER2	In vitro	[180]

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
