# Peer review of "Antibody Conjugation of Nanoparticles as Therapeutics for Breast Cancer Treatment"

_ijms, 2020, doi:10.3390/ijms21176018_

Round 1

Reviewer 1 Report

The manuscript "Antibody Conjugation of Nanoparticles as..." is a repetition of the old theme that targeting by antibody conjugation to nanoparticles is risky and not satisfactory although the highly selective principle of antigen-antibody recognition is available. Many publication refer to reviews and methodical descriptions often concerned with in vitro investigations. The given information is not forthcoming to rigorously attack the problem why the clinics sparsely or not at all use the antibody conjugated nanoparticles. Before questioning degrees of producing commercially cheap production, the knowledge of underlying mechanisms to achieve best nanomedicines has to be proposed. In this regard, a juxtaposition with non-antibody targeting is missing. Further a critical account of so many parameters important in the design of targeted NPs should be centered in the report, because they have been well discussed and published, but here only mentioned insufficiently and at the side of the manuscript. These are e.g. size, shape, degradation, dynamics of receptor attachment, halo formation, release of NPs from their biological receptors during targeting (thermodynamic as well as kinetic aspects especially of high affinity antigen-antibody binding), rates of vascular excit into tumor tissue compared to rates of renal, liver, etc. clearance. In this context the nanodrug underlying geometric spacing of poly valency (avidity) opposite to spatial incompatibilities or the overcrowding of targeting moieties on the NP surface has to be deeper than reported here, and other considerations In the present form, the review is to a great part only descriptive and collecting numbers of FDA clinical approval but not proposing a way out by encouraging changes on cellular, microenvironmental, molecular, chemical, physical-chemical, and tissue levels.

Author Response

The purpose of the reported work was to give an overview of antibody conjugated nanoparticles (ACNPs) for those interested in the use of these nanomedicines for the treatment of cancer. The authors chose to work on a review manuscript which might be conceptually descriptive to collect all the information about the use of ACNPs in breast cancer therapy. We covered inorganic NPs, polymeric NPs and immunoliposomes as the most studied targeted nanocarriers. Outcomes from in vitro and in vivo studies are therefore discussed in different sections (3.1-3.3) by continually comparing data with non-targeted NPs. Unfortunately, there is very scarce literature in this indication particularly for in vivo studies using ACNPs. Outlook and recent implications in breast cancer therapy is discussed at the end of the manuscript once the main theme is deeply reviewed.

As the generation and assessment of ACNPs are quite multidisciplinary, we considered that IJMS were the most appropriate journal to disseminate our work. To contextualize the discussion for a wide scope of readership, the authors started giving basic concepts about targeted nanomedicines. For that purpose, in a first section an introduction regarding antibody drug conjugates as the most successful targeted delivery systems in clinic is providing. In this context, it is important to understand for the readers the concept of conjugation. Later on, a section discussing about the pros and cons of ACNPS in comparison to ADCs and NPs for clinical translation is included. As we have previously mentioned, these sections try to contextualize the state of the art and at no point is the key goal of this review. As the referee mentioned, the critical parameters in the design of targeted NPs have been very well discussed and published elsewhere.  

Following referee suggestions and to improve the quality of our revision, we have decided to incorporate in our work a section to cover the critical key points ACNPs must face to reach the clinic. In this extensive section we have described limitations as well as options for improvement. We do consider that this new part improves substantially the manuscript, providing the reader with all the elements for a critical evaluation of the current data.  

We really appreciate the comments for the referee, but we would like to clarify that the purpose of our report is not to propose new strategies to overcome the limitations of ACNPs for the treatment of cancer. The objective was to review outcomes of ACNPs in breast cancer therapy

Reviewer 2 Report

The review by Juan et al focuses on the ADCs and their application in the treatment of breast cancer. The manuscript is well organized and scientifically sound. I would just ask the authors to include a paragraph and maybe a figure describing in general how ADCs and their variants work (binding, internalization, intracellular trafficking, drug release).

Author Response

A paragrah concerning to the mechanism of action of the ACNPs is included in the revised manuscript (Section 2.1). Furthermore, a new figure 3 intend to help discussion.

Round 2

Reviewer 1 Report

The authors have understood the problems of the manuscript and defend their point of view. Nevertheless more detailed opinion could have added.

If the editor agrees with their primary intention to summarize the global efforts on the theme and with their now added hurried criticism on the current accomplishment in the field, then I like to approve the manuscript.

Eggehard Holler